# OpenReview forum: "Topological Complexity of Reasoning Chains in Large Language Models: Persistent Homology of Attention Manifolds"
_mathai.club/MathAI/2026/Conference — MathAI 2026 Conference Submission_

### Official Review · Reviewer_8SSm · 2026-03-10
**interesting paper**

**Rating:** 6
**Confidence:** 2

**Review:**

this is an interesting paper and has a lot of promise.

I like the ideas and scope of the paper.

I would suggest to the author(s) to remove jargons and help simplify the paper. For example, there are terminologies like persistent homology and nerve theorem. However there is no explanation for this.

I would like to see an improved version of this. I am also not an expert on this topic so I have to defer to someone who has more expertise in this.

also a figure is missing in the paper.

---

### Official Review · Reviewer_rXfD · 2026-03-13
**Topological Complexity of Reasoning Chains in Large Language Models: Persistent Homology of Attention Manifolds ()A Review)**

**Rating:** 7
**Confidence:** 2

**Review:**

The authors introduce a topological framework for analyzing and predicting the reasoning capabilities of large language models. By constructing simplicial complexes from attention weight matrices across transformer layers, they compute
persistent homology invariants  that capture the structural complexity of information flow during chain-of-thought reasoning. The
main theoretical contribution is a Topological Reasoning Capacity Theorem: for a transformer with L layers and H attention heads, the maximum reasoning chain length it can faithfully represent is bounded by the total persistence.

The key insight — that reasoning capacity is fundamentally constrained by the topological complexity (persistent homology) of attention manifolds — opens a new research direction at the intersection of topology, deep learning, and cognitive science.The main theoretical contribution, the Topological Reasoning Capacity Theorem, provides the first mathematically rigorous bound on the length of reasoning chains that a transformer can faithfully perform.

---

### Decision · Program_Chairs · 2026-03-20

**Decision:**

Reject

**Comment:**

After careful evaluation by the Program Committee, we regret to inform you that your submission has not been accepted for presentation at MathAI 2026.

All submissions underwent a rigorous two-stage review process. Unfortunately, the reviewers identified one or more of the following concerns with your paper:

- Insufficient mathematical rigor or novelty relative to the existing body of work in the field;
- Presentation of results that substantially overlap with or rephrase previously published findings without clear original contribution;
- Significant issues with technical quality, including but not limited to broken or non-existent references, unsupported claims, or methodological gaps;
- Indications that the manuscript may have been generated with the assistance of large language models without substantial original intellectual contribution by the authors.

We received a large number of submissions this year, and the selection process was highly competitive. We encourage you to carefully consider the reviewers’ feedback (available through OpenReview), revise your work accordingly, and consider submitting an improved version to a future edition of MathAI or to another appropriate venue.

We appreciate your interest in MathAI and hope you will continue to engage with the conference community.

With kind regards,

MathAI 2026 Program Committee
International Conference on Mathematics of Artificial Intelligence
https://mathai.club
OpenReview: https://openreview.net/group?id=mathai.club/MathAI/2026/Conference
MathAI Telegram: https://t.me/MathAI_club
IAIC International AI Committee: https://t.me/iaic_world
Email: mathai.club@yandex.ru